# Lung Ultrasound Is Often, but Not Always, Normal in Healthy Subjects: Considerations for COVID-19 Pandemic

**DOI:** 10.3390/diagnostics11010082

**Published:** 2021-01-06

**Authors:** Alberto Raiteri, Margherita Alvisi, Ilaria Serio, Federico Stefanini, Francesco Tovoli, Fabio Piscaglia

**Affiliations:** 1Department of Medical and Surgical Sciences, University of Bologna, 40138 Bologna, Italy; margherita.alvisi@studio.unibo.it (M.A.); Ilaria.serio@aosp.bo.it (I.S.); francesco.tovoli2@unibo.it (F.T.); fabio.piscaglia@unibo.it (F.P.); 2Division of Internal Medicine, IRCCS Azienda Ospedaliero-Universitaria di Bologna, via Albertoni 15, 40138 Bologna, Italy; federico.stefanini@aosp.bo.it

**Keywords:** lung ultrasound, point-of-care ultrasound, subpleural consolidations, interstitial syndrome, COVID-19, healthy individuals

## Abstract

Background: Lung ultrasound (LU) is becoming an increasingly important diagnostic tool in detecting lung involvement in Corona Virus Disease 2019 (COVID-19). The aim of this study was to ascertain the likelihood of finding LU abnormalities; mimicking lung involvement; in COVID-19 negative healthy individuals. Methods: We performed LU on 265 healthcare workers; not presenting COVID-19 major symptoms and in good health; during the course of a serological screening program for COVID-19 in our General Hospital. LU results were reported as total Lung Ultrasound Score (LUS) using a 12-zone method of reporting. Results: 250/265 subjects were included in the COVID-19 negative group. LU was not completely normal (LUS ≠ 0) in 65/250 COVID-19 negative subjects (26%) and in 12/15 (80%) poorly symptomatic COVID-19 positive subjects; with a multifocal pattern in 12.7% vs. 66.7% of cases respectively. Age and COVID-19 positivity were independent predictors of total LUS. A total LUS ≥ 2 had a sensitivity of 66.67% and a specificity of 85.60% in detecting COVID-19 positivity. Conclusions: A slightly altered LU can be quite frequent in healthy COVID-19 negative subjects. LU can have a role in confirming but not screening COVID-19 poorly symptomatic cases.

## 1. Introduction

Lung ultrasound (LU). LU is a simple and safe methodology for differential diagnosis of lung disorders and for dyspnoea work-up [1]. In particular, LU is accurate in differentiating “wet” causes (cardiogenic pulmonary oedema and acute inflammation) from “dry" causes (chronic obstructive pulmonary disease and asthma) of acute dyspnoea.

LU proved to be a valid alternative to computed tomography (CT) scan and a better choice than chest X-ray (CXR) to detect lung involvement in Corona Virus Disease 2019 (COVID-19) [2,3,4]. This is not only due to its diagnostic performances, but also to its wide availability and potential use at the patient’s bedside, in emergency department and at the patient’s home [4].

Mono-focal or multi-focal irregular pleural thickening, small subpleural consolidations or group of B-lines, were reported to be indicative of pulmonary involvement in COVID-19 mild to moderate symptomatic cases (Figure 1). This was the case during the peak of COVID-19 pandemic. What it is not well known is the prevalence of these findings in a healthy COVID-19 negative general population. This knowledge is becoming of even greater importance after the first peak of pandemic, in order to avoid misdiagnoses in a situation of endemic disease or of second wave. Prevalence of these findings in non COVID-19 infected patients must be known in order to ascertain the diagnostic role of LU in case of low-moderate pre-test prevalence of the disease. Moreover, specificity and sensitivity of these findings in asymptomatic/pauci-symptomatic, COVID-19 positive, subjects are not well described.

When detecting abnormalities at LU, physicians are prone to associate them to the current clinical suspicion under investigation. Examples of this situation are described in the literature [5]. Clinicians often assume that no LU alteration is to be found in the lung of apparently healthy subjects. Whether this assumption is correct or not has never been extensively investigated. Therefore, when an abnormality is found at LU, it is impossible to know which is the likelihood that such abnormality was present from before the current active disease. A better understanding of how a LU alteration is predictive of an actual disease is essential to avoid misinterpretations. This is much truer in the era of COVID-19 pandemic, when LU is often used as first screening tool in a large number of suspected cases. LU is part of the Eco-FAST (Focused Assessment with Sonography for Trauma) and POCUS (Point-of-Care Ultra Sonography), and sometimes used as an extension of an “other-site” ultrasound exam, especially in Emergency Departments and Internal Medicine wards.

The aim of this study was to determine the prevalence and the characteristics of LU alterations in a COVID-19 negative healthy general population. The secondary endpoint was to determine the sensitivity and specificity of LU alterations amongst asymptomatic/pauci-symptomatic COVID-19 positive subjects.

## 2. Materials and Methods

The study was conducted at Policlinico Sant’Orsola-Malpighi University Hospital (Bologna, Italy) during a three-week period.

A LU was offered to all healthcare workers (HCW) interested in participating in the study.

The choice of enrolling HCW was obviously imposed by the impossibility to open the study to common citizens during COVID-19 pandemic.

Exclusion criteria included the presence of COVID-19 major symptoms (fever, cough, dyspnoea, myalgias, fatigue, expectoration, anosmia) in the 7 days prior to the exam. These symptoms are the most evocative ones in COVID-19, described in more than 30% of patients [6]. An additional exclusion criterion was the presence of ≥2 severe comorbidities among pulmonary, cardiologic, endocrine-metabolic, gastroenteric, neurological, and active neoplastic disease.

A total of 265 volunteer healthcare workers (HCW) (79 men, 186 women) were enrolled. They included 72 medical doctors, 146 nurses, and 47 health practitioners, from 9 different either COVID or non-COVID units. The median age was 37 (IQR 27.5/46.5).

Every subject gave written informed consent before entering the study. The study was conducted in accordance with the Declaration of Helsinki, and the protocol was approved by the Ethics Committee of Area Vasta Emilia Centro, Regione Emilia Romagna (CE-AVEC) (Project identification code 638/2020/Sper/AOUBo, 17 June 2020).

Minor symptoms, comorbidities, smoking habits, and number of working days in a COVID-19 ward were also recorded. Minor symptoms, suffered at least once in the 7 days prior to the investigation, included: headache, sore throat, rhinorrhea, non-cardiac chest pain, diarrhea, nausea and/or vomiting, abdominal pain. Presence of pulmonary, cardiovascular endocrinological, gastrointestinal, neurological, neoplastic, and other comorbidities were recorded. Working days in a COVID-19 ward were counted until the date of the LU.

Every participant underwent serology test (ST) and/or nasopharyngeal swab (NPS) real-time polymerase chain reaction (RT-PCR) test during the same week of the ultrasound exam (Median −1 days, IQR −4 + 2). These tests were already planned and independently performed by the Occupational Medicine Unit as part of the HCW screening program. The program scheduled the performance of ST every two weeks for every HCW, at the time of the present study. According to the occupational medicine screening protocol, an NPS was performed in case of positive ST (IgM+ and or IgG+). The study was conducted at the time of the first part-session of screening program. Accordingly, when we refer to first ST, we specifically refer to the first overall ST performed by the subject. Some of the subjects included in the study underwent NPS regardless of the result of first ST, because considered to have possibly been in contact with a COVID-19 case, according to Occupational Medicine Unit procedures. All NPS considered in this study were performed after no more than 7 days following the LU.

According to the current evidences about COVID-19 ST interpretation, subjects were divided into two groups [7]. The first group, named “COVID-19 positive”, included subjects in whom first ST resulted IgM+/IgG− or IgM+/IgG+, and/or NPS resulted positive, followed by a consistent second ST (IgM+/IgG−, IgM+/IgG+ or IgM−/IgG+). This pattern of ST is evocative of acute or subacute COVID-19 infection. Subjects with first ST resulted IgM−/IgG+ were not included in this group because they were consistent with a previous, and non-strictly recent, COVID-19 infection. Furthermore, a complete recovery from COVID-19 LU alterations is described in most of the patients after healing. Therefore, non-strictly recently infected subjects (first ST IgM−/IgG+) were included in the “COVID-19 negative” group. The second group, named “COVID-19 negative” included subjects in whom both first and second ST resulted IgM-/IgG. It also included subjects in whom second ST resulted negative (IgM−/IgG−) despite a first positive ST (not confirmed cases), and subjects IgM−/IgG+ both first and second ST as previous explained, since indicating a past, non-recent, infection. All subjects in this group must not have complained of any new major symptom of COVID-19 in the 2 weeks following LU, in order to further confirm their negativity.

LU were performed with Esaote MyLab 70 XVG and Esaote MyLab 7, with a 2.5–7.5 MHz convex probe. LU scans were carried out in sitting position. Ultrasounds were performed by either of two MDs, experts in LU, R.A. or A.M. Both operators were residents in Internal Medicine, with personal experience in wards dedicated to COVID-19 patients. The investigators performed each LU blindly to the results of the ST or NPS test of all investigated subjects, either because ST was still to be performed or to be reported at the time of the US scan, or they were not kept informed of the results. All contact precautions aiming at avoiding COVID-19 spread were put in place. Subjects were investigated wearing face masks and while sitting with their backs to the operator, in order to further limit droplets spread. Ultrasound probe and equipment were sanitized after every use.

In order to test the inter-operator reliability, the exam was repeated by the two operators independently from each other, within 48 h, in 15 volunteers.

For LU, we used a 12-zone method of reporting, in whom lung is divided in 12 areas (Figure 2). We started from the ICU lung ultrasound score (LUS) reporting system [8]. We decided to add the presence of thickened, irregular pleural line, or small subpleural consolidations (<10 mm) to the score, giving it a score of 1. All other LU alterations (LUA), were preserved as described in the original LUS score, receiving a score increased by 1 (+1).

The adopted scoring system was therefore as follows:

Score 0: Pulmonary A line pattern, surmounted by a regular, narrow pleural line.Score 1: Thickened, irregular pleural line, or small subpleural consolidations (<10 mm), almost constantly associated with uneven B lines (up to 3 per scanning field of view).Score 2: B lines, more than three per scanning field of view, but occupying < 50% of the scanning field of view.Score 3: Confluent B lines > 50% of the scanning field of view.Score 4: Consolidations > 10 mm, with or without bronchogram.

Presence of pneumothorax or pleural effusion were recorded separately. Every lung zone was examined in longitudinal and transversal axis.

Statistical analysis was performed with JASP, JASP Team (2020). JASP (Version 0.12.2, computer software, University of Amsterdam, Amsterdam, The Netherlands). Continuous variables were expressed as median and interquartile range. Categorical variables were expressed as frequencies and percentages. Variables which presented a *p* < 0.150 at the univariate analysis were included in multivariate analysis. A *p* < 0.05 was considered as the cut-off for statistical significance.

## 3. Results

### 3.1. Patients Characteristics

A total of 183 out of 265 enrolled subjects (69%) were completely free of symptoms. 31% of subjects presented one or more minor symptoms (Table 1). A total of 58 out of 265 subjects (22%) presented one or more minor comorbidity and 78% had no comorbidity (Table 1). 

169/265 (63.8%) participants were non-smokers, 14 (5.3%) ex-smokers and 82 (30.9%) active smokers. 

Median of working days in a COVID-19 ward was 20 days (IQR 9.5–29.5). 

### 3.2. Serology Test and Nasopharyngeal Swab Results

In 254/265 subjects first and second serology test (ST) provided the same pattern (either positive or negative) with no change over time (Table 2). A change in the serology pattern between the first and second serology test was instead documented in the remaining 11 subjects (Table 2).

Nasopharyngeal swab (NPS) was found positive in one case with positive serology at first ST and in two subjects who converted from first negative ST to a positive one.

Five cases with positive serology at first test, but totally negative at second serology test and with negative NPS were considered false positive and included in the COVID19 negative group. Also, two subjects who were found IgM-/IgG+ at both first and second ST were included in the COVID19 negative group, as indicative of a past non recent infection. Accordingly, the COVID-19 positive group consisted of 15 subjects and the COVID19 negative group of 250.

### 3.3. Lung Ultrasound Results

In the COVID-19 negative group, 74% of the subjects (185/250) presented a normal lung ultrasound appearance (LUS = 0), but 26% (65/250) showed some abnormalities. However, the vast majority (81.5%, 53/65 subjects) suffered only mild abnormalities: total lung ultrasound score either 1 or 2 was present respectively in in 29 and 24 subjects and none had LUS ≥ 5 (Figure 3).

In the COVID-19 positive group, despite subjects were not at all or poorly symptomatic, the rate of normal lung ultrasound appearance markedly dropped, to 20% (3/15). 80% (12/15 subjects) showed some abnormalities: LUS 1 was present in 2 subjects, (13.3%), LUS 2 in 6 subjects (40%), LUS 3 in 1 subject (6.7%), LUS 4 in 2 subjects (13.3%) and LUS 5 in 1 subject (6.7%) (Figure 3).

None of the subjects in either group presented pleural effusions or pneumothorax.

Since the total lung score does not express the extension of the lung involvement (as it is affected also by its severity) we analysed the number of areas affected by lung ultrasound alterations (LUA), regardless of their individual severity.

As reported above, a total of 65/250 COVID-19 negative subjects (26%) showed at least one affected area. Abnormalities were limited to only one area in around half of those (33/65, 50.7%), whereas two or more affected lung areas were present in the remaining half (32/65, 49.3%).

The number of affected areas was significantly greater in the 12/15 (80%) COVID-19 positive subjects showing LU abnormalities. In particular, 10/15 of all COVID-19 positive subjects showed abnormalities in ≥2 lung areas (67.7% vs. 12.8% in COVID-19 negative subjects). Abnormalities limited to one lung area were seen in only 16.7% (2/12) of COVID-19 positive subjects (vs. 50.7% in COVID-19 negative subjects).

No subject presented any lung area with a severity of abnormality scored 3 or 4, as these are likely to be expected only in symptomatic patients. A LUA 1 (thickened, irregular pleural line, or small subpleural consolidations < 10 mm) was found in ≥1 lung area in 62/250 (24.8%) COVID-19 negative healthy subjects (Figure 4). LUA 1 was instead present in one or more lung areas in 7/12 (67.7%) of COVID-19 positive patients (Figure 4).

LUA 2 (B lines, more than three per scan, but occupying < 50% of the field of view) was found in one or more lung ultrasound areas in 12/250 (4.8%) COVID-19 negative, expectedly healthy subjects (Figure 5). In 8 out of 12 of these subjects, LUA 2 was associated with a LUA 1 in another lung ultrasound area.

In total, LUA 2 was found in ≥1 lung ultrasound area in 4/15 (26.7%) COVID-19 positive subjects (Figure 5). In 2 out of 4 of these subjects, it was associated with an abnormality scored as 1 in another lung area.

In COVID-19 negative group, expectedly healthy subjects, no correlation was found between the presence of LUA 1 or LUA 2 and the presence of lung comorbidities (χ^2^: 0.895, *p* = 0.344 and χ^2^: 7.281-5, *p* = 0.993 respectively), sex (χ^2^: 1.748, *p* = 0.186 and χ^2^: 0.174, *p* = 0.677 respectively), and smoking habits (χ^2^: 1.706, *p* = 0.192 and χ^2^: 0.541, *p* = 0.462 respectively). No correlation was found between the presence of LUA1 and the presence of minor symptoms (χ^2^: 4.124, *p* = 0.344). A positive correlation was found between the presence of LUA 2 and the presence of minor symptoms (χ^2^: 10.542, *p* = 0.032).

No difference was found in mean age between subjects presenting or not LUA 1 (t: −1.498, *p* = 0.135). A statistical difference was found in mean age between subjects presenting or not LUA 2: 45.6 ± 2.3 years (LUA 2 present) vs. 38.7 ± 0.7 years (LUA 2 absent), (t: −2.497, *p* = 0.013).

### 3.4. Involvement of Individual LU Areas According to COVID-19 Status

A correlation network analysis was performed in order to visually estimate the relationship between specific lung areas and COVID-19 positivity. LU alterations mostly correlated to COVID-19 positivity were detected in right and left upper-posterior and lower-lateral lung areas (Figure 1 and Table 3).

### 3.5. Predictors of Altered LU

We verified the presence of predictors of LU alterations (total LUS) amongst COVID-19 negative subjects, assessing the role of age, sex, smoking habit (active or previous), COVID-19 minor symptoms, comorbidities, and pulmonary comorbidities. At univariate analysis age, smoking habits, rhinorrhea and nausea/vomiting reached a *p* < 0.150 and were included in the model. A multiple linear regression showed that age and nausea/vomiting were independently correlated with the total LUS and could predict it with the following regression equation: LUS = 0.066 + (0.010 × Age) + (1.038 × nausea/vomiting (F = 4.791, *p* = 0.002) (Table 4).

### 3.6. Predictors of COVID-19 Positivity

We then verified the possible predictors of COVID positivity, assessing the role of total LUS, age, sex, number of working days in a COVID-19 ward, presence of minor symptoms, comorbidities, and active smoking habit.

The univariate analysis showed a correlation between COVID-19 positivity and total LUS (*p* > 0.001), age (*p* = 0.134), diarrhea as minor symptom (*p* = 0.067), number of working days in a COVID-19 ward (*p* = 0.149) and active smoke habit (*p* = 0.147).

The logistic regression model, including total LUS, age, diarrhea, number of working days in a COVID-19 ward, and active smoking habit resulted statistically significant, χ^2^ (259) = 30.806, *p* < 0.001, McFadden R^2^ = 0.267, with a sensitivity of 26.7% and a specificity of 99.2%. Considering a COVID-19 positivity prevalence of 5.6% in the population studied, Positive Likelihood Ratio resulted 22.22 (95% CI 5.46–90.42) and Negative Likelihood Ratio resulted 0.74 (95% CI 0.55–1.01).

Analysing regression coefficients (Table 5), increasing total LUS was independently associated with an increased likelihood of COVID-19 positivity with an OR of 2.6 (*p* < 0.001). The number of days worked in a COVID-19 ward was independently associated with an increased likelihood of COVID-19 positivity with an OR of 1.063 (*p* = 0.03).

The relationships between total LUS, or between days worked in a COVID-19 ward, and COVID-positivity appeared both to be logarithmic as shown by the graphs in Figure 6.

### 3.7. Total LUS as Predictor of COVID-19 Positivity

A ROC curve analysis of the correlation between total LUS and COVID-19 positivity, identified total LUS score of 2 as the best cut-off in order to maximize sensitivity and specificity (AUC 0.761).

A chi-square (χ^2^) test of association, between the presence of LUS score ≥ 2 and COVID-19 positivity, was performed to ascertain sensitivity and specificity of LUS at this selected cut-off.

There was a significant association between LUS ≥ 2 and COVID-19 positivity, with a likelihood ratio of 19.434 (*p* < 0.001). Sensitivity 66.67% (95% CI 38.38–88.18%), Specificity 85.60% (95% CI 80.63–89.71%), Positive Predictive Value 21.74% (95% CI 14.81–30.73%), Negative Predictive Value 97.72% (95% CI 95.43–98.87%). Considering a prevalence of COVID-19 positivity of 5.6% in the population studied (which is well in keeping with the expected prevalence of actual or past recent COVID-19 infections in healthcare workers in a high incidence area), Positive Likelihood Ratio resulted as 4.63 (95% CI 2.90–7.40) and a Negative Likelihood Ratio resulted as 0.39 (95% CI 0.19–0.80).

### 3.8. Reproducibility Analysis

An inter-rater agreement was assessed on 15 pairs of observations to determine reproducibility among raters. Subjects were selected according to the availability to repeat the exam within 48 h. The two raters were blind about the results of each other. Agreement was assessed on the basis of total LUS score and individual lung ultrasound areas scores. The inter-rater reliability for the total LUS was found to be substantial, with Cohen’s kappa of 0.793 (*p* < 0.001) (Table 6). The inter-rater reliability for areas scored 1 and 2 was found to be substantial, with Cohen’s kappa of 0.770 (*p* < 0.001) and 0.707 (*p* < 0.001) respectively (Table 7 and Table 8).

## 4. Discussion

The present study showed that a not negligible portion (approximately 30%) of expectedly healthy COVID-19 negative subjects presented some lung ultrasound abnormalities. This information is relevant at the time of usage of lung ultrasound as a screening tool for lung involvement in COVID-19 suspected or confirmed patients. However, the vast majority of abnormalities in COVID-19 negative subjects were not severe (score 1 of our reporting system, corresponding to thickened, irregular pleural line, or small subpleural consolidations < 10 mm) and were found in 24.8% of them. Alterations scored 2, corresponding to more than three B-lines per scan, but occupying less than 50% of the field of view, were found in only 4.8% of COVID-19 negative subjects. The result was significantly different from COVID-19 positive subjects, who showed much higher rates. Additionally, the number of involved lung areas was more limited in the COVID-19 negative population than in COVID-19 positive one, despite the paucity or absence of symptoms.

The presence of the alteration scored 2 (LUA 2) was significantly more prevalent in subjects with COVID-19 atypical symptoms amongst COVID-19 negative healthy subjects. Since such symptoms are non-specific, it cannot be excluded that these subjects suffered from an unrecognized pathological condition other than COVID-19. To summarize, LUA 2 is not only much rarer in healthy subjects than LUA 1 but also less frequent in totally asymptomatic subjects.

Given these premises we cannot anymore assume that any lung ultrasound abnormality can be attributed to a COVID-19 infection both at the time of COVID-19 pandemic peak or in endemic conditions. We therefore tried to dissect whether any additional use of ultrasound parameters could help to distinguish COVID-19 positive from negative subjects.

Using the total LUS as a scale, there was a 2.6-fold increase in the likelihood of COVID-19 positivity by every point increase in total LUS score. The Positive Likelihood Ratio was high (22.22, 95% CI 5.46–90.42). However, Sensitivity was as low as 26.7%, making total LUS score inefficient if thought as a screening tool in patients without significant symptoms.

Consistently with the finding that some lung ultrasound abnormalities were found up to 30% of expectedly healthy subjects (COVID-19 negative), the Negative Likelihood Ratio did not result low enough (0.74, 95% CI 0.55–1.01) to allow the use of a total LUS of 0 to rule out COVID-19 positivity. In other words, a Negative Likely Ratio of this level does not significantly increase the post-test probability to be COVID-19 negative. Obviously, a normal lung ultrasound still maintains a role, as a LUS score 0 reassures the physician that, even if the patient would be COVID-19 positive, the lung involvement would anyhow be non-significant, if any.

Using total LUS of 2 as a cut-off for predicting COVID-19 positivity, sensitivity was 66.7% and specificity 85.6%. The Negative Predictive Value remained very high (97.7) with a Negative Likelihood Ratio of 0.39 (95% CI 0.19–0.80). Thus, the probability to be COVID-19 negative in case of a LUS < 2 is 97.7%, 2.56 times higher than the pre-test probability. The Positive Likelihood Ratio (4.63) remained moderately high, meaning a 4.63-fold increasing in pre-test probability to be COVID-19 positive in case of a LUS ≥ 2.

In COVID-19 negative healthy subjects incidental ultrasound lung alterations seem to be more frequent in low-posterior lung areas bilaterally (9.6% right and 8% left), bringing the rates to levels not too dissimilar to those of COVID-19 positive subjects (13.3% right and 6.7% left). Conversely, poorly or not symptomatic COVID-19 positive subjects tended to show an involvement of the upper-posterior lung areas (33% on the right and 20% on the left), differently from COVID-19 subjects in which this occurrence was rare (3.2% both on the right and on the left). The preferential involvement of the upper posterior areas in COVID-19 is consistent with what is described empirically in current literature [3].

In our model, the increase in age corresponded to a very small increase of the total LUS (linear regression coefficient of 0.10). In general, we can assume that in younger patients, the presence of lung ultrasound alterations is more suggestive of an underling pathology, whereas in elder patient it may be caused by a previous, currently recovered, disease.

The low sensitivity of lung ultrasound for the diagnosis of COVID-19 is not unique among diagnostic tests, all of which suffer some problems related to the risk of infection transmission. [9]. CT scan has demonstrated a higher sensitivity than RT-PCR NPS in detecting COVID-19 cases, while chest X-rays showed a lower sensitivity than RT-PCR tests [3,10]. Moreover, chest X-ray can be held as conclusive only in more striking cases. It is worth noting that a single RT-PCR NPS is not a perfect diagnostic tool, since it suffers from a low sensitivity, slightly greater than 70%. To summarize, clinical, epidemiological, and CT features compatible with COVID-19 could lead to its diagnosis despite a negative RT-PCR test [3], notwithstanding caution is always recommended. However, the use of CT scan, which would be preferable, is limited by its lack of portability and radiation exposure. In this context, lung ultrasound maintains its utility, given its ease of use and portability, even at the bedside at patients’ home, which is out of reach for CT and chest X-rays. The present study brought evidence to further improve lung ultrasound use, especially in poorly or not symptomatic subjects, like those that are usually investigated in a primary care setting.

To conclude, we found that a total LUS ≥ 2 is to be held as the best cut-off to justify further investigations (i.e., repetition of a negative ST or performance of NPS, or another imaging technique) in adults presenting with minor or no symptoms. Additionally, the involvement of the upper posterior lung areas should also raise a strong suspicion upon COVID-19. Conversely, a LUS < 2 can help supporting the exclusion of COVID-19 together with the clinical context.

The finding of LUS 1 remains a clinical challenge as it can be found in a non-negligible number of COVID-19-negative expectedly healthy subjects.

## 5. Limitations

The absence of a gold standard exam for diagnosing COVID-19 poses some limitations in defining cases and controls with certainty.

Enrolling healthy subjects amongst common citizens, letting them circulate further in a healthcare facility, was deemed ethically unacceptable during COVID-19 pandemic.

## Figures and Tables

**Figure 1 diagnostics-11-00082-f001:**
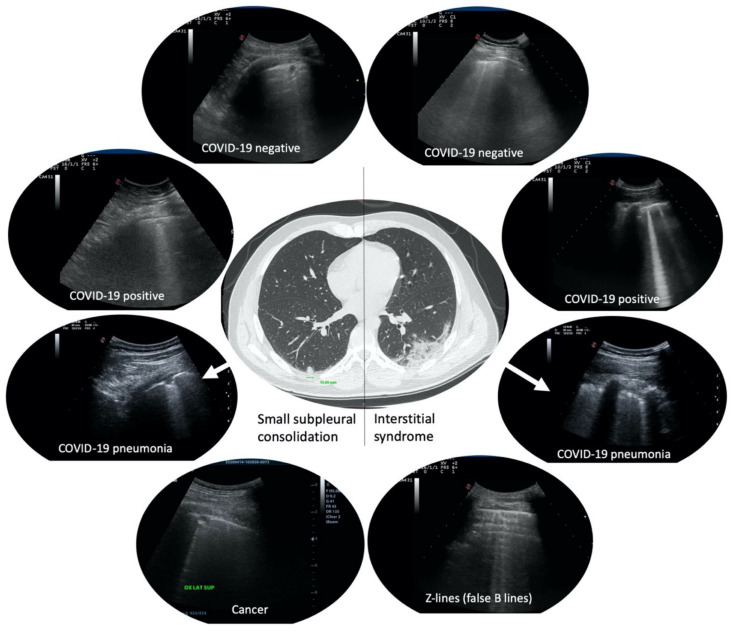
Examples of small subpleural consolidations and group of B-lines more than three per field of scan. In the center, a computer tomography scan of a real COVID-19 patient presenting a small subpleural consolidation in the right lung (left of the image) and a ground-glass area in the left lung (right of the image), and their corresponding features in lung ultrasound (LU) (arrows). On the left, small subpleural consolidations in four different patients with different diagnoses. On the right, group of B-lines more than three per field of scan in four different patients with different diagnoses.

**Figure 2 diagnostics-11-00082-f002:**
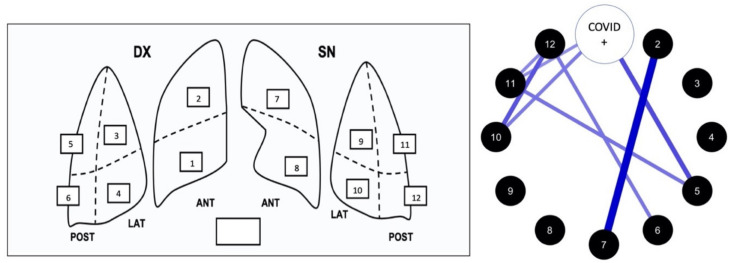
On the **left**: scheme of LUS score system of report with the 12 lung areas. Total lung score (0 to 48) is obtained from the sum of the score of each single lung area (0 to 4). On the **right**: correlation network analysis of the relationship between alterations of single area and COVID-19 positivity (see “Results” for more details).

**Figure 3 diagnostics-11-00082-f003:**
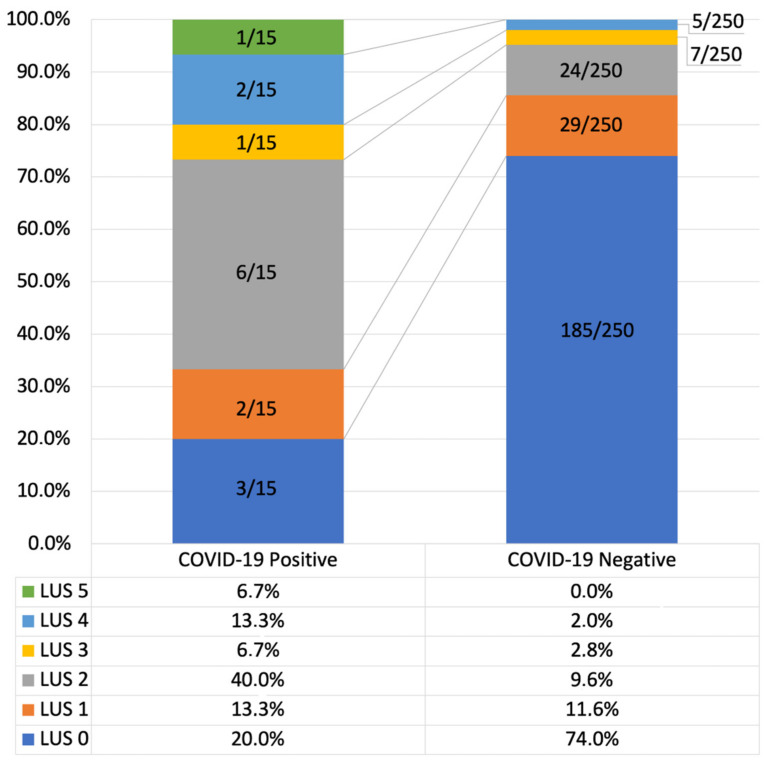
Total Lung Ultrasound Score (LUS) in COVID-19 positive and negative subjects.

**Figure 4 diagnostics-11-00082-f004:**
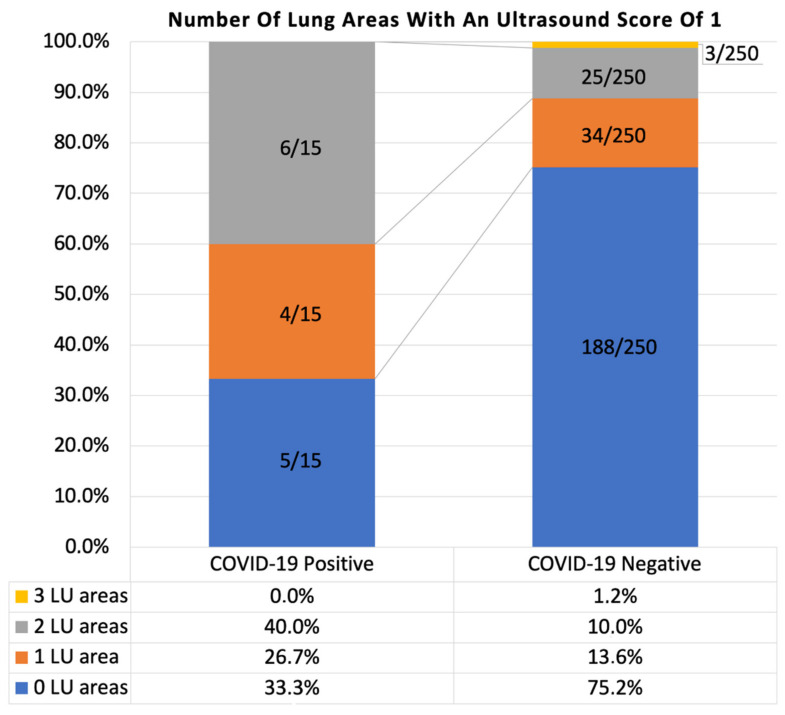
Number of lung ultrasound areas interested by an alteration scored as 1 (thickened, irregular pleural line, or small subpleural consolidations < 10 mm) in COVID-19 positive and negative groups. LU = lung ultrasound.

**Figure 5 diagnostics-11-00082-f005:**
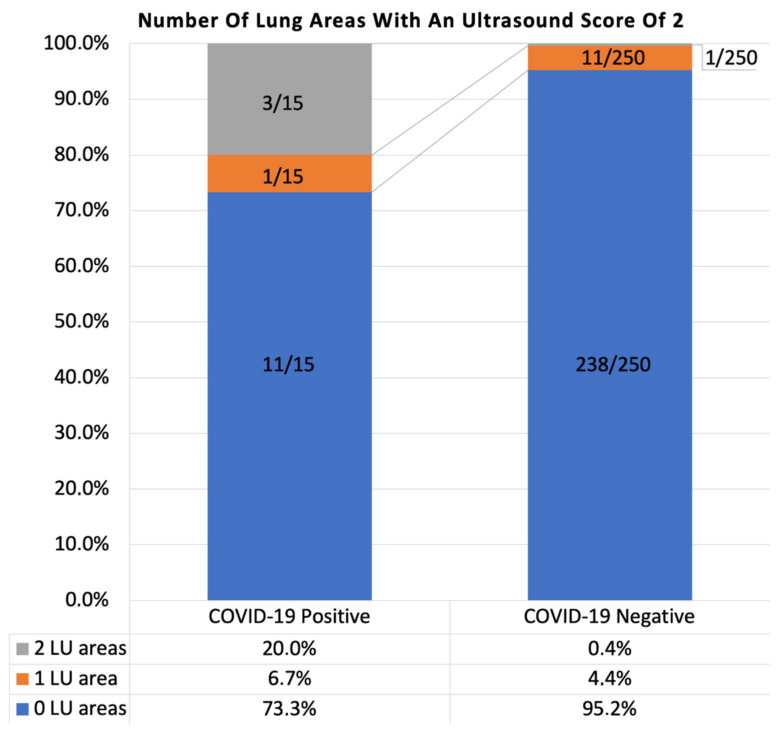
Number of lung areas interested by lung alteration 2 (LUA 2) in COVID-19 positive and negative group. LU = lung ultrasound.

**Figure 6 diagnostics-11-00082-f006:**
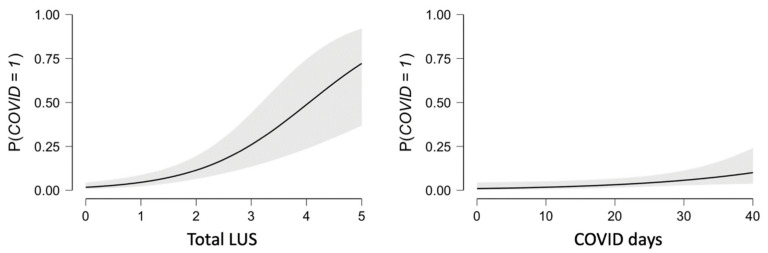
Logarithmic relationship between total LUS and COVID-19 positivity, and days worked in a COVID-19 ward (COVID days) and COVID-19 positivity.; COVID 0 = negative, COVID 1 = positive.

**Table 1 diagnostics-11-00082-t001:** Prevalence of symptoms and comorbidities amongst enrolled subjects.

Symptoms	%	Comorbidities	%
Headache	6.4	Pulmonary	8.6 (20/23 asthma)
Sore throat	17.7	Cardiovascular	4.9 (10/13 hypertension)
Rhinorrhea	6.0	Endocrinological	7.9 (21/21 hypothyroidism)
Non-cardiac chest pain	1.1	Gastrointestinal	1.5
Diarrhea	3.7	Neurological	1.1
Nausea and/or vomiting	2.3	Neoplastic	1.1 (3/3 free of disease for more than 5 years)
Abdominal pain	0.75	Other	2.6 (5/7 allergic rhinitis)
Others	1.1		

**Table 2 diagnostics-11-00082-t002:** First and second serology test (ST) results. * indicates subjects in which the NPS was found positive (one case for each asterisk). IgM = immunoglobulin M, IgG = immunoglobulin G.

**FIRST ST**	**SECOND ST**					
		**IGM−/IGG−(2ST)**	**IGM+/IGG+**	**IGM+/IGG−**	**IGM−/IGG+**	**SUM**
	**IGM−/IGG−**	243	1 *	0	1 *	245
	**IGM+/IGG+**	0	5	0	3 *	8
	**IGM+/IGG−**	3	0	4	1 *	8
	**IGM−/IGG+**	2	0	0	2	4
	**SUM**	248	6	4	7	265

**Table 3 diagnostics-11-00082-t003:** Individual lung areas involvement in COVID-19 positive and negative subjects. R = right, L = left.

Lung Areas	COVID-19 Positive	COVID-19 Negative
R upper-anterior	13.3%	1.6%
R lower-anterior	0%	0%
R upper-lateral	20%	5.2%
R lower-lateral	13.3%	2.4%
R upper-posterior	33.3%	3.2%
R lower-posterior	6.7%	9.6%
L upper-anterior	6.7%,	1.2%
L lower-anterior	0%	0.4%
L upper-lateral	6.7%,	2.8%
L lower-lateral	20%	4.4%
L upper-posterior	20%	3.2%
L lower-posterior	13.3%	8%

**Table 4 diagnostics-11-00082-t004:** Linear regression coefficients of predictors of total LUS in COVID-19 negative subjects.

Coefficients					
Model	Unstandardized	Standard Error	Standardized	t	*p*
(Intercept)	0.005	0.218		0.024	0.98
Age	0.010	0.005	0.120	1.944	0.05
Smoke	0.101	0.062	0.101	1.635	0.10
Rhinorrhea	−0.405	0.240	−0.104	−1.686	0.09
Nausea	1.040	0.372	0.173	2.795	0.006

**Table 5 diagnostics-11-00082-t005:** Logistic regression coefficients of predictors of COVID-19 positivity.

Coefficients							
					Wald Test		
	Estimate	Standard Error	Odds Ratio	z	Wald Statistic	df	*p*
(Intercept)	−5.814	1.542	0.003	−3.770	14.212	1	<0.001
Total LUS	0.955	0.218	2.600	4.384	19.216	1	<0.001
Age	0.021	0.030	1.021	0.719	0.516	1	0.472
COVID days	0.061	0.028	1.063	2.166	4.691	1	0.030
Diarrhea (1)	0.967	0.962	2.630	1.005	1.011	1	0.315
Active Smoke (1)	−1.177	0.815	0.308	−1.444	2.086	1	0.149

**Table 6 diagnostics-11-00082-t006:** Confusion matrix verifying inter-rater agreement. From the top, total LUS rated by operator 1 and 2.

Reproducibility Analysis Total LUS					
	Operator 1				
Operator 2	0	1	2	3	Total
0	7	0	0	0	7
1	1	4	0	0	5
2	0	0	2	1	3
3	0	0	0	0	0
Total	8	4	2	1	15

**Table 7 diagnostics-11-00082-t007:** Number of lung ultrasound areas interested by lung ultrasound alteration 1 (LUA1) found by operator 1 and 2.

Reproducibility Analysis LUA 1				
	Operator 1			
Operator 2	0	1	2	Total
0	7	1	0	8
1	1	4	0	5
2	0	0	2	2
Total	8	5	2	15

**Table 8 diagnostics-11-00082-t008:** Number of lung ultrasound areas interested by lung ultrasound alteration 2 (LUA 2) found by operator 1 and 2.

Reproducibility Analysis LUA 2			
	Operator 1		
Operator 2	0	1	Total
0	14	0	14
1	0	1	1
Total	14	1	15

## Data Availability

The data presented in this study are available on request from the corresponding author. The data are not publicly available due to privacy policy.

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
