# Peer review of "Lung Ultrasound Is Often, but Not Always, Normal in Healthy Subjects: Considerations for COVID-19 Pandemic"

_diagnostics, 2021, doi:10.3390/diagnostics11010082_

Round 1
Reviewer 1 Report
The paper is an interesting analysis performed in the setting of academic hospital performing lung ultrasound in health care workers. The analysis identify high prevalence of subclinical abnormalities on lung ultrasound. The main disappointing part is the discussion section where no reference is reported to support the main finding of the paper and thus need to be extensively re-written.Author Response
Please see the attachment

Reviewer 2 Report
Overall, congratulations on a very interesting, practical and critical study. I believe the article raises a very important point regarding the "weaknesses" and "strengths" of LUS in COVID-19 diagnosis.
My only reservation concerns the presentation of the results and the structure of conclusions and discussions.
- Much data is repeated in figures and in the text. I propose to select the old key for the final conclusions, describe them in the text and indicate that the detailed results are included in subsequent figures or described in tables.
- The discussion as it stands is basically a repetition of the results. In the same time, the conclusions take the form of discussions. In my opinion, these parts need to be rewritten. I suggest referring to other studies on COVID-19 imaging in the discussion. Referring to the strengths and weaknesses of CT, clinical symptoms etc.
- Conclusions should be less elaborate in literary form. The place to discuss the results is discussion. Applications should be brief, more in the form of "take home massages"
Summing up, I believe that the work should be published, but please re-edit the indicated parts of the article firstly so that it would be more legible and understandable also for people who do not perform LUS
Round 2
Reviewer 2 Report
Dear Authors
Congratulations on your interesting work, thank you also for including my suggestions in the revision of the manuscript. I am satisfied with the amendments made. I am pleased to recommend the publication of this article
Best regards